# Proteomic characterization of acute kidney injury in patients hospitalized with SARS-CoV2 infection

Ishan Paranjpe[1,24], Pushkala Jayaraman [2,24], Chen-Yang Su [3,4], Sirui Zhou [3,5], Steven Chen[6], Ryan Thompson [2,7], Diane Marie Del Valle[7], Ephraim Kenigsberg [6,8,9], Shan Zhao [10], Suraj Jaladanki [7,11], Kumardeep Chaudhary[12], Steven Ascolillo [7], Akhil Vaid [7], Edgar Gonzalez-Kozlova [7], Justin Kauffman[7], Arvind Kumar [7], Manish Paranjpe[13], Ross O. Hagan[7,11], Samir Kamat[7], Faris F. Gulamali[7,11], Hui Xie[14], Joceyln Harris[14], Manishkumar Patel [14], Kimberly Argueta[14], Craig Batchelor[14], Kai Nie[14], Sergio Dellepiane[15], Leisha Scott[14], Matthew A. Levin [7,10], John Cijiang He [15], Mayte Suarez-Farinas [16], Steven G. Coca[15], Lili Chan[15], Evren U. Azeloglu [15], Eric Schadt [8], Noam Beckmann [7,8], Sacha Gnjatic [17], Miram Merad [6], Seunghee Kim-Schulze[6,14], Brent Richards [3,18,19,20], Benjamin S. Glicksberg [21], Alexander W. Charney [3,8,22,23] & Girish N. Nadkarni [2,5,11,15 ✉]

## Abstract

**Background** Acute kidney injury (AKI) is a known complication of COVID-19 and is associated with an increased risk of in-hospital mortality. Unbiased proteomics using biological specimens can lead to improved risk stratification and discover pathophysiological mechanisms.

**Methods** Using measurements of ~4000 plasma proteins in two cohorts of patients hospitalized with COVID-19, we discovered and validated markers of COVID-associated AKI (stage 2 or 3) and long-term kidney dysfunction. In the discovery cohort ($N = 437$), we identified 413 higher plasma abundances of protein targets and 30 lower plasma abundances of protein targets associated with COVID-AKI (adjusted $p < 0.05$). Of these, 62 proteins were validated in an external cohort ($p < 0.05$, $N = 261$).

**Results** We demonstrate that COVID-AKI is associated with increased markers of tubular injury (NGAL) and myocardial injury. Using estimated glomerular filtration (eGFR) measurements taken after discharge, we also find that 25 of the 62 AKI-associated proteins are significantly associated with decreased post-discharge eGFR (adjusted $p < 0.05$). Proteins most strongly associated with decreased post-discharge eGFR included desmocollin-2, trefoil factor 3, transmembrane emp24 domain-containing protein 10, and cystatin-C indicating tubular dysfunction and injury.

**Conclusions** Using clinical and proteomic data, our results suggest that while both acute and long-term COVID-associated kidney dysfunction are associated with markers of tubular dysfunction, AKI is driven by a largely multifactorial process involving hemodynamic instability and myocardial damage.

## Plain language summary

Acute kidney injury (AKI) is a sudden, sometimes fatal, episode of kidney failure or damage. It is a known complication of COVID-19, albeit through unclear mechanisms. COVID-19 is also associated with kidney dysfunction in the long term, or chronic kidney disease (CKD). There is a need to better understand which patients with COVID-19 are at risk of AKI or CKD. We measure levels of several thousand proteins in the blood of hospitalized COVID-19 patients. We discover and validate sets of proteins associated with severe AKI and CKD in these patients. The markers identified suggest that kidney injury in COVID-19 patients involves damage to kidney cells that reabsorb fluid from urine and reduced blood flow to the heart, causing damage to heart muscles. Our findings might help clinicians to predict kidney injury in patients with COVID-19, and to understand its mechanisms.

---

A full list of author affiliations appears at the end of the paper.

Severe acute respiratory syndrome coronavirus 2 (SARS-CoV-2) is a novel coronavirus that has caused the coronavirus disease 2019 (COVID-19) pandemic. Although effective vaccines are available, novel variants that may evade neutralizing antibodies exist in the population and have led to high case counts and periodic case surges. COVID-19 most commonly presents with fever, cough, and dyspnea[1,2] and is associated with acute respiratory distress syndrome (ARDS). However, the clinical syndrome resulting from SARS-CoV-2 infection is broad, ranging from asymptomatic infection to severe disease with extrapulmonary manifestations[3], including acute kidney injury[4], acute myocardial injury[5,6] and thrombotic complications[7–11]. The CRIT-COV-U research group in Germany recently developed a urinary proteomics panel COV50 that could consider this variability in infection by generating biomarkers that can indicate adverse COVID-19 outcomes based on the WHO severity scale[12].

Acute kidney injury (AKI) is a particularly prominent complication. The rates of AKI vary greatly based on patient population, but evidence suggests that at least 30% of hospitalized patients and 50% of patients in the intensive care unit (ICU) develop AKI[1,4,13–16]. Although the rate of AKI in hospitalized COVID-19 patients has decreased since the initial surge in 2020, the incidence remains high[17]. Like community-acquired pneumonia[18], AKI is increasingly recognized as a common complication of COVID-19 in the hospitalized setting and confers significantly increased morbidity and mortality[19].

There is a limited understanding of the pathophysiology of COVID-19-associated AKI. A recent paper[20] compared transcriptomics and proteomics of postmortem kidney samples of patients with severe COVID-19 and autopsy-derived control cohorts of sepsis-AKI and non-sepsis-AKI. The work found common inflammatory pathways and regulatory responses including the downregulation of oxidative signaling pathways between COVID-19 AKI and sepsis-AKI. They also confirmed the observation of tubular injury in almost all their COVID-19 AKI samples while drawing similarities between the inflammation response of sepsis-associated AKI and COVID-19 associated AKI. Histopathological reports from autopsy specimens have provided conflicting insights into the pathological changes in the kidney in COVID-19. A report of 26 patients who died with COVID-19 AKI revealed acute tubular injury as a prominent mechanism[21]. Additionally, the presence of viral particles in the tubular epithelium and podocytes in autopsy specimens has been reported[21,22], which is evidence of direct viral invasion of the kidney. In addition, coagulopathy and endothelial dysfunction are hallmarks of COVID-19[23] and may also contribute to AKI. Finally, SARS-CoV-2 may directly activate the complement system[24]. In addition to these mechanisms, systemic effects of critical illness (hypovolemia, mechanical ventilation) and derangements in cardiac function and volume may also contribute to COVID-19 AKI.

In addition to morbidity and mortality in the acute setting, COVID-19 is also associated with long term manifestations i.e., the post-acute sequelae of SARS-CoV2 (PASC)[25]. Kidney function decline is a major component of PASC and a study of >1 million individuals found that survivors of COVID-19 had an elevated risk of post-acute eGFR decline[26], suggesting long term kidney dysfunction may occur following the acute infection.

Given the high incidence of COVID-19 associated kidney dysfunction, the unknown pathophysiology, and the urgent need for better approaches for risk stratification for long term kidney function decline we aimed to characterize the proteomic changes associated with COVID associated AKI and long-term kidney function. Proteomic biomarkers have previously shown success in predicting COVID-19 outcomes[27–29]. Other work[12] has also applied urinary proteomic profiling to predict worsening of COVID-19 at early stages of the infection. Prior research using minimally invasive proteomics assays supports the use of peripheral serum as a readily accessible source of proteins that accurately reflect the human disease state[30–33]. We measured protein expression of >4000 proteins from serum samples collected in a diverse large cohort of hospitalized patients with COVID-19 and validated significant results in an independent cohort and identified proteins that are significantly different between patients with and without AKI. We then determined whether these proteomic perturbations also characterize post-discharge kidney function decline as measured by estimated glomerular filtration rate (eGFR).

Our results demonstrate that COVID-AKI is associated with increased markers of tubular injury (NGAL) and myocardial injury. Using estimated glomerular filtration (eGFR) measurements taken after discharge, we find that 25 of the 62 AKI-associated proteins are significantly associated with decreased post-discharge eGFR (adjusted $p < 0.05$). Notably, these include desmocollin-2, trefoil factor 3, transmembrane emp24 domain-containing protein 10, and cystatin-C indicating tubular dysfunction and injury.

## Methods

**Patient cohort**. An overview of the discovery cohort selection process is provided in Fig. 1. We prospectively enrolled patients hospitalized with COVID-19 between March 24, and August 26, 2020, at five hospitals of a large urban, academic hospital system in New York City, NY into a cohort as previously described[34]. The cohort enrolled patients (with informed consent) who were admitted to the health care system with a COVID-19 infection and had broad inclusion criteria without specific exclusion criteria. The Mount Sinai Institutional Review Board approved this study under a regulatory approval allowing for access to patient-level data and biospecimen collection[35]. This research was reviewed and approved by the Icahn School of Medicine at Mount Sinai Program for the Protection of Human Subjects (PPHS) under study number 20-00341. Peripheral blood specimens were collected at various points during the hospital admission for each patient. Data for the analysis and the clinical data covariates are available in Synapse syn35874390 found here. Access to the data and steps to process the clinical information to create the cohort is detailed on the site.

The validation cohort included a prospective biobank from Quebec, Canada that enrolled patients hospitalized with COVID-19, as previously described[29]. Validation cohort was The Biobanque Québécoise de la COVID-19 (BQC19) cohort. For individual-level data in BQC19, BQC19 received ethical approval from the Jewish General Hospital research ethics board (2020-2137) and the Centre hospitalier de l'Université de Montréal institutional ethics board (MP-02-2020-8929, 19.389). All participants gave informed consent. More information on the plasma proteome from BQC19 can be viewed at https://www.mcgill.ca/genepi/mcg-covid-19-biobank. Access to the data of BQC19 can be obtained upon approval of requests via bqc19.ca. Patients were recruited from the Jewish General Hospital and Centre Hospitalier de l'Université de Montréal. Peripheral blood specimens were collected at multiple time points after admission.

We defined an AKI cohort using proteomic data acquired at the last available timepoint during the hospital course for all individuals. Patients who developed AKI after the last specimen collection timepoint were excluded. Controls were defined as individuals who developed AKI stage 1 or did not develop AKI during their hospital course.

**a** **Prospective cohort of patients enrolled between Mar 2020 – Aug 2020**

**b** **Timeline of measurements taken**

**Fig. 1 Overview of the discovery cohort selection process. a** Cohort selection strategy overview. (*abbreviation legend*: AKI Acute Kidney Injury, ESKD End Stage Kidney Disease, Ctrls Controls) **b** Estimated Glomerular Filtration Rate (eGFR) measurements recorded post-discharge for returning patients until 12/21/2021. Source data for **a** are generated from Supplementary Data 3, which is in turn generated from the clinical data table in the Synapse data repository syn35874390.

**Serum collection and processing**. Blood samples were collected in Serum Separation Tubes (SST) with a polymer gel for serum separation as previously described[35]. Samples were centrifuged at $1200 g$ for 10 min at 20 °C. After centrifugation, serum was pipetted to a 15 mL conical tube. Serum was then aliquoted into cryovials and stored at −80 °C.

**Definition of acute kidney injury**. We defined AKI (stage 2 or 3) as per Kidney Disease Improving Global Outcomes (KDIGO) criteria: an increase in serum creatinine of at least 2.0 times the baseline creatinine[36]. For patients with previous serum creatinine measurement available in the 365 days prior to admission, the minimum value in this period was considered the baseline creatinine. For patients without a baseline creatinine in this period, a baseline value was calculated based on an estimated glomerular filtration rate (eGFR) of 75 ml/min per 1.73 m[2] as per the KDIGO AKI guidelines.

**Clinical data collection**. We collected demographic and laboratory data collected as part of standard medical care from an institutional electronic health record (EHR) database. We defined clinical comorbidities using diagnostic codes recorded in the EHR before the current hospital admission. To account for disease severity at the time of specimen collection, we defined supplemental oxygen requirement as 0 if the patient was not receiving

supplemental oxygenation or on nasal cannula, 1 if the patient was receiving non-invasive mechanical ventilation (CPAP, BIPAP), or 2 if the person was receiving invasive mechanical ventilation.

**Somalogic proteomic assay**. We used the *SomaScan* discovery platform to quantify levels of protein expression[37]. The *SomaScan* platform is a highly multiplexed aptamer based proteomic assay based on Slow Off-rate Modified single-stranded DNA Aptamers (SOMAmers) capable of simultaneously detecting 4497 proteins in biological samples in the form of relative fluorescent units (RFUs). The assay was run using the standard 12 hybridization normalization control sequences to assess for variability in the Agilent plate quantification process, five human calibrator control pooled replicates, and 3 quality control pooled replicates to control for batch effects. Standard preprocessing protocols were applied as per Somalogic's guidelines published previously[37] The specificity and stability of the SOMAScan assay has been described previously[38]. Briefly, the data was first normalized using the 12 hybridization controls to remove hybridization variation within a run. Then, median signal normalization is performed with calibrator samples across plates to remove variation in sample-to-sample differences attributable to variations due to pipetting, reagent concentrations, assay timings and other technical aspects. Data was then calibrated to remove assay

differences between runs. Standard Somalogic acceptance criteria for quality control metrics were used (plate scale factor between 0.4 and 2.5 and 85% of QC ratios between 0.8 and 1.2). Samples with intrinsic issues such as reddish appearance or low sample volume were also removed as part of the Somalogic quality control protocol. After quality control and normalization procedures, the resulting relative fluorescence unit (RFU) values were log2 transformed.

**Dimensionality reduction.** Principal component analysis (PCA) was performed using log2 transformed RFU values of all proteins. Pairwise plots of the top three principal components were plotted.

**Differential expression analysis for prevalent AKI.** Using data from the AKI cohort, $\log_2$ transformed normalized protein values were modeled using multivariable linear regression in the Limma framework[39] Models were adjusted for age, sex, history of chronic kidney disease (CKD), and supplemental oxygen requirement (0,1, or 2 [see above]) at the time of specimen collection. P-values were adjusted using the Benjamin-Hochberg procedure to control the false discovery rate (FDR) at 5%. FDR was performed on discovery and validation data with respect to all proteins measured.

**Proteomic characterization of long-term kidney function in discovery cohort.** Outpatient creatinine values measured after discharge were used to compute estimated glomerular filtration rate (eGFR) values the CKD-EPI equation. All values were taken from the EHR as part of routine clinical care with follow-up until 12/2/2021. To determine whether AKI associated protein expression correlated with post-discharge kidney function, we fit a mixed effects linear regression model with random intercept. Using the discovery cohort, protein expression of AKI-associated proteins measured at the last available timepoint during admission was used. The dependent variable was eGFR and the model was adjusted for age, sex, baseline creatinine, history of CKD, maximum AKI stage during the hospital admission, and day of eGFR measurement after hospital discharge. Models included a random effect of patient ID to adjust for correlation between eGFR values taken from the same individuals. Significance was evaluated using a t-test with Satterthwaite degrees of freedom implemented in the *lmerTest* R package[40]. P-values were adjusted using the Benjamin-Hochberg procedure to control the false discovery rate (FDR) at 5%.

We then plotted the post-discharge eGFR values over time for individuals separated by protein expression tertiles (bottom 33rd percentile, middle 33rd percentile, and top 33rd percentile). We transformed data using the LOESS smoothing function as implemented in the *ggplot* R package.

**Data analysis and visualization.** We performed all statistical analysis using R version 4.0.3. Protein–protein interaction (PPI) network was constructed using the Network X package in Python v3.4.10 to display a Minimum Spanning Tree (MST) using Prim's algorithm. Network clustering was conducted using the MCL cluster algorithm and functional enrichment was carried out using the STRING[41] database in Cytoscape[42]. Using results from a recent publication[30], we also annotated our signature AKI proteins with reported protein quantitative trait loci (pQTLs) for the set of COVID AKI-associated proteins. For each AKI-associated protein, we determined whether cis and trans pQTL associations had been reported.

**Reporting summary.** Further information on research design is available in the Nature Portfolio Reporting Summary linked to this article.

## Results

**Discovery and validation cohort overview.** To discover proteins associated with COVID-AKI, we enrolled a prospective cohort of patients hospitalized with COVID-19 admitted between March 24, 2020 and August 26, 2020 into a biobank as previously described[34]. Cases were defined as patients who developed AKI (stage 2 or 3) during their hospital admission and controls included all other patients (Fig. 1). Characteristics of cases and controls in the discovery cohort are provided in Supplementary Data 1 (sheet "Table 1"). Patients who developed AKI (stage 2 or 3) had a greater prevalence of diabetes (42% vs 22%, $p < 0.001$), and chronic kidney disease (31% vs 5%, $p < 0.001$) and more frequently required intubation (46% vs 11%, $p < 0.001$). Patients who developed AKI (stage 2 or 3) also had a significantly lower minimum systolic blood pressure (104 vs 110, $p < 0.001$), greater maximum pulse (106 vs 94, $p < 0.001$), white blood cell count (12.9 vs 8.8, $p < 0.001$), ferritin (2210 vs 1030 $p < 0.001$), and frequency of vasopressor use (48% vs 14%, $p < 0.001$). We validated proteomic associations in an external cohort from Quebec, Canada. Characteristics of the validation cohort are provided in Supplementary Table 1 (See Supplementary Information). In the validation cohort, compared to controls, AKI (stage 2 or 3) cases have a significantly higher prevalence of CKD (29% vs 11%, $p = 0.01$) and a higher rate of intubation at the time of sample collection (49% vs 13%). FDR was performed on discovery and validation data with respect to all proteins measured.

**Identification of proteins associated with prevalent AKI.** In the discovery cohort, serum levels of 4496 proteins were quantified using the *SomaScan* platform using samples collected at multiple time points during the hospital course (Supplementary Table 2, See Supplementary Information) as previously described[34]. We first identified proteins associated with prevalent AKI using measurements taken after the onset of AKI in cases and the last available measurement in controls (Fig. 1, 71 cases and 366 controls). The top three principal components (PCs) distinctly separate samples by case status (Fig. 2a). We fit a multivariable linear regression model for the log2 normalized protein expressions adjusted for age, sex, history of chronic kidney disease (CKD), and maximum oxygen requirement at the time of blood draw. We identified 413 proteins with higher plasma abundances and 30 proteins with lower plasma abundance (Supplementary Data 2).

**Validation of AKI-associated proteins.** We then performed an external validation of AKI-associated proteins in a prospective biobank cohort from Quebec, Canada. 443 proteins in the discovery cohort are significantly associated with AKI (FDR adjusted $P < 0.05$) while 71 proteins in the validation cohort are significantly associated with AKI ($p < 0.05$). Of the proteins significantly associated with AKI in the discovery cohort, 62 are also associated with AKI in the validation cohort ($p < 0.05$), See Supplementary Data 1 (sheet "Table 2"). The hypergeometric test for overlap between the significant proteins in the validation and discovery cohorts was also significant ($P = 2.133E-159$). Additionally, the Cohen's Kappa between the two protein lists based on P-value was 0.501. All validated proteins associate with an increased risk of AKI with nominal significance. The correlation of fold changes of validated proteins in the discovery and validation cohort show a Pearson correlation score of 0.71 (Fig. 2b). The 62-protein signature distinctly separate AKI cases from

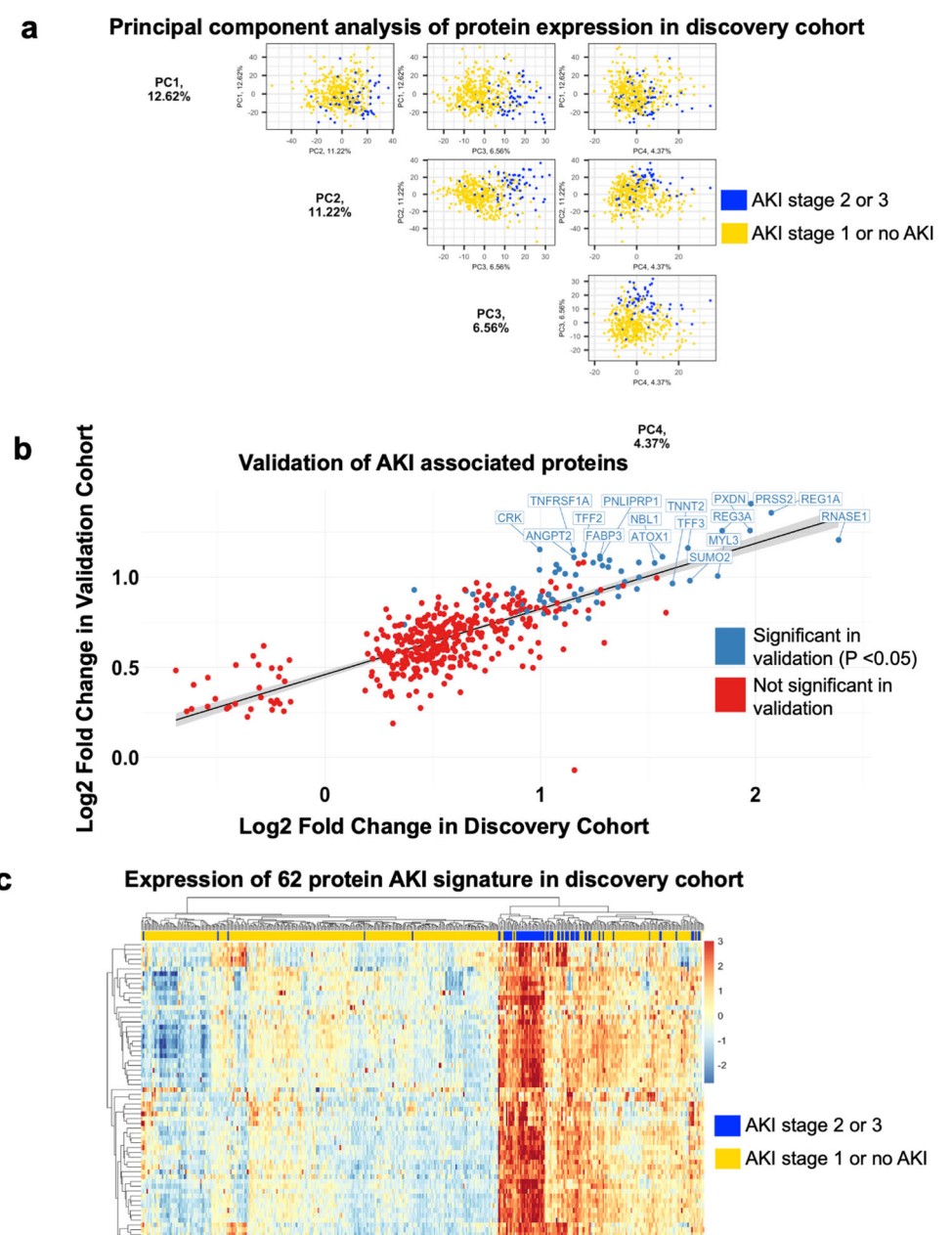

**Fig. 2 Analysis of the clinical and proteomic data. a** Top 3 Principal Components show separation of the sample by Acute Kidney Injury (AKI) (stage 2 or 3) case status. **b** External validation of AKI associated proteins in the discovery cohort shows high correlation with increased risk of AKI with significance of p < 0.05. (n = 62). The black line of regression runs diagonally across, and the gray shaded area represents the region wthin 95% Confidence Intervals around the regression line. **c** Expression heatmap shows a distinct separation of the cases and controls using the 62 significant proteins identified from the validation cohort in the discovery cohort. Source data for the PCA plots in **a** are found in the Synapse data repository syn35874390. Source Data for **b** are provided in Supplementary Data 1(sheet "Table 2"). Source Data for **c** is from the raw data matrix provided in syn35874390 but filtered for the 62 proteins documented in the sheet "Table 2" in Supplementary Data 1.

cohorts in the discovery cohort (Fig. 2c). To assess how many of our candidate proteins had orthogonal evidence for target specificity, we sought to identify how many of our proteins contained reported plasma protein quantitative trait loci (pQTL) associations from a recent publication by Ferkingstad et al. (Nat Genet, 2021). Of the 62 AKI associated proteins, 45 have both cis and trans pQTLS, 14 have only trans pQTLs, and 2 had cis pQTLs (See Supplementary Fig. 1, Supplementary Information). Protein–protein interaction (PPI) network analysis reveal enrichment of several highly connected proteins, including LCN2 (alternative name: NGAL), REG3A, and MB (Fig. 3a). The AKI-associated protein network also includes a cluster of cardiac

structural proteins (Fig. 3b), TNNT2, TTN, MYL3, SRL, and NPPB (alternative name: BNP).

**Proteomic characterization of post-acute kidney dysfunction**. Given the previously reported association of COVID-19 AKI with long-term eGFR decline[43], we hypothesize that significant proteomic markers associated with COVID-19 AKI are also associated with post-discharge eGFR. We included all outpatient eGFR measurements taken after discharge from patients in the Mount Sinai Biobank cohort of the 437 patients in the cohort, 181 patients had at least one outpatient post-discharge eGFR

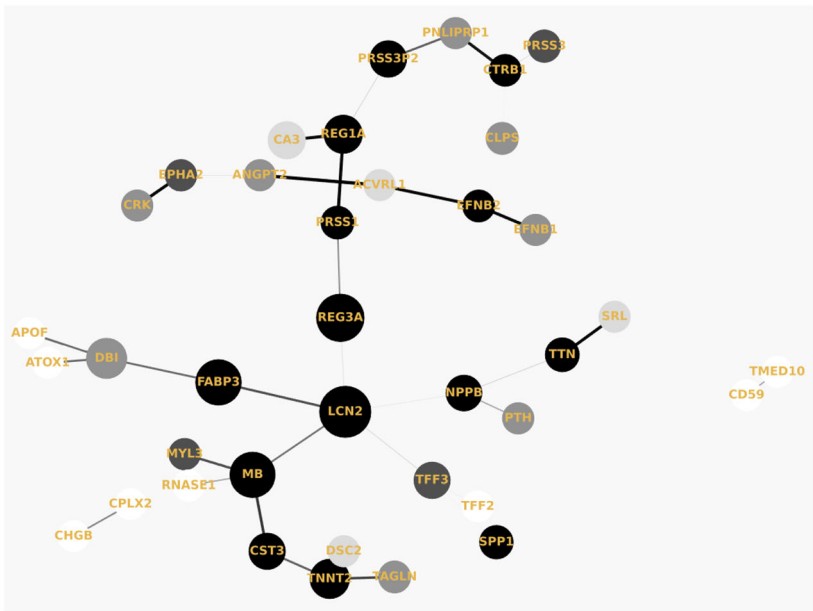

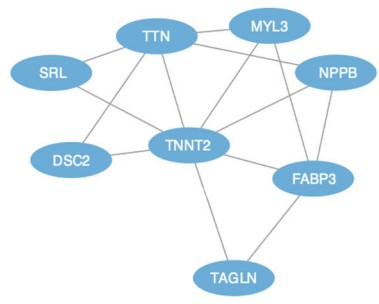

**Fig. 3 Protein–protein interaction (PPI) and clustering analysis for functional annotation of the 62 differentially expressed proteins. a** Protein–protein interaction (PPI) network (Minimum Spanning Tree) of the 62 overlapping AKI associated proteins with a score >0.4. The size of each node corresponds to number of interactions and the thickness of the edges represent the weight of the interactions between the nodes. **b** Markov Clustering Algorithm (MCL) algorithm was used to identify tightly connected cluster of proteins which was functionally enriched for cardiac structure proteins using the STRING (https://string-db.org/) database. Source data for Fig. 3a. AKI-associated protein–protein interaction network on our GitHub repository.

measurement. The median number of eGFR measurements was 4 with an interquartile of 9. The first post-discharge eGFR was measured at a median of 37 days after discharge. The last post-discharge eGFR was measured at a median of 374 days after discharge (See Supplementary Fig. 2, Supplementary Information).

We used a mixed effects linear model accounting for baseline creatinine, AKI stage during the COVID admission and repeated eGFR measurements to associate the 62 protein AKI signature with long-term eGFR. Of the 62 AKI-associated proteins, 25 are significantly (FDR adjusted $P < 0.05$) associated with long-term post-discharge eGFR (Figs. 4 and 5a). All 25 eGFR-associated proteins are negatively correlated with post-discharge eGFR (See Supplementary Data 2). However, the strength of association with AKI is not significantly associated with the strength of association with post-discharge eGFR. Proteins most strongly associated (by $P$-value) with decreased post-discharge eGFR include desmocol-lin-2, trefoil factor 3, transmembrane emp24 domain-containing protein 10, and cystatin-C (Fig. 5b).

## Discussion

Using proteomic profiling in two large groups of patients hospitalized with COVID-19, we report several observations. First,

we identified specific protein markers of AKI and post-discharge kidney dysfunction, both well-documented sequelae of COVID-19[4,43]. Second, in the acute phase, tubular injury and hemody-namic perturbation may play a role. Thus, characterization of the peripheral blood suggests specific large-scale perturbations of the proteome that accompany both AKI and long-term eGFR decline with implications for more specific prognostic models and tar-geted therapeutic development.

Based on our results, we hypothesize that COVID AKI may involve several mechanisms: tubular injury, neutrophil activation, and hemodynamic perturbation. First, we found significantly higher plasma abundances of NGAL (LCN2), a canonical marker of tubular injury that is also involved in neutrophil activation. NGAL is secreted by circulating neutrophils and kidney tubular epithelium in response to systemic inflammation or ischemia. Since renal tubular epithelial cells express the angiotensin-converting enzyme 2 (ACE2) receptor which enables SARS-CoV2 viral entry into cells, direct tubular infection may cause the release of NGAL into the serum and urine. This potential mechanism is supported by our results and remains a testable hypothesis. Although NGAL is a known marker for intrinsic AKI accompanied by tubular injury, it is relatively insensitive to pre-renal AKI caused by hemodynamic disturbance[44,45]. However,

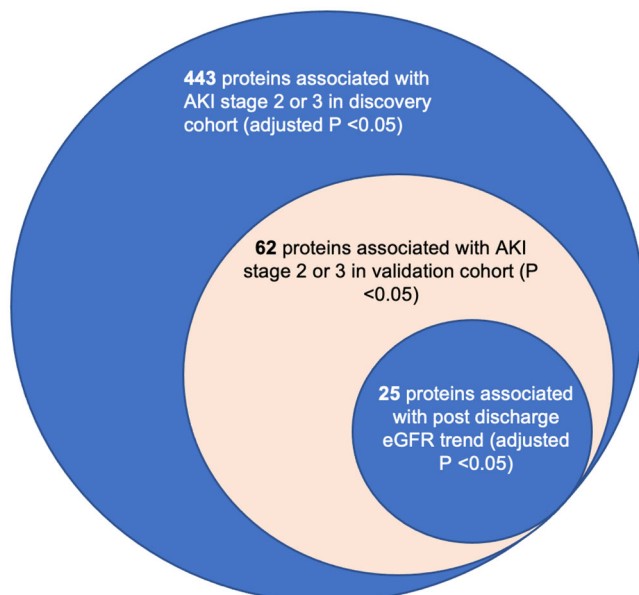

**Fig. 4 Nested Venn diagram of the analyses performed.** The Venn diagram showing the count of significant genes at each analysis step. Of the 443 proteins found to be significantly associated with Acute Kidney Injury (AKI) stage 2 or 3 in the discovery cohort, 62 were significantly associated with AKI stage 2 or 3 in the validation cohort. Of the 62 AKI-associated proteins, 25 are significantly (FDR adjusted $P < 0.05$) associated with long-term post-discharge estimated glomerular filtration rate (eGFR). Source data for Fig. 4. Results Overview is acquired from Supplementary Data 2 and Supplementary Data 1(sheet "Table 3").

our results demonstrate higher plasma abundance of BNP, a protein released in the setting of volume overload as well as several cardiac structural proteins (cardiac troponin T, titin, myosin light chain 1, and sarcalumenin). This proteomic signature may represent either myocardial injury leading to decreased renal perfusion or impaired filtration by the kidney in response to injury. Myocardial injury has been previously reported in patients hospitalized with COVID-19[6] and thus may contribute to the multifactorial nature of COVID-AKI. It is worth noting that in addition to myocardial injury, BNP may also be increased in critical illness due to pro-inflammatory cytokine release.

Since COVID-AKI increases the risk of long-term eGFR decline[43], we then sought to determine whether these two phenomena shared common proteomic markers. Surprisingly, we found that although almost half of the AKI-associated proteins were also significantly associated with post-discharge eGFR decline, the strengths of associations were not correlated. While COVID-AKI is likely caused by a combination of intrinsic tubular injury and hemodynamic disturbance in the setting of critical illness, long term eGFR decline was associated with increased expression of trefoil factor 3 (TFF3), a known prognostic marker for incident CKD[46]. Trefoil factors are a class of small peptides expressed in colonic and urinary tract epithelia that play essential roles in regeneration and repair of epithelial tissue[47,48]. Immunohistochemistry reveals TFF3 expression is localized to the tubular epithelial cells in kidney specimens from patients with CKD[46], suggesting that long term eGFR decline may be associated with renal tubular epithelial damage. The exact pathological role of TFF3 in the renal tubules is unclear but it has been hypothesized to play a role in repair of kidney damage[49]. Additionally, TFF3 release from the renal interstitium has also been hypothesized to direct the epithelial-to-mesenchymal transition

(EMT) in renal interstitial fibrosis, a main pathway that leads to ESKD[46]. Our results implicate tubular damage in both AKI and long term eGFR decline suggesting that SARS-CoV2 may preferentially target this region of the nephron. While AKI in the acute setting may be a result of ischemia and decreased renal perfusion associated with critical illness, the specific elevation of TFF3 associated with eGFR decline implicates a more general pattern of tubular injury that underlies COVID mediated kidney dysfunction. Since the ACE2 is preferentially expressed in the tubular epithelial cells of the kidney[50,51], the elevation of markers of tubular damage in the plasma may represent direct viral invasion of tubular epithelia cells. However, again this would need to be tested using biopsy/autopsy specimens or other mechanistic studies. Direct viral entry into the kidney remains controversial and using our current data we are not able to comment on this mechanism.

Our study should be interpreted in the context of certain limitations. First, samples were collected during the hospital course of patients with confirmed COVID-19. However, the time points were not systematic due to logistical challenges during the peak of the COVID-19 pandemic and thus are not standardized between patients. Since a subset of patients had AKI at the time of admission, these patients were excluded from our analysis since specimens were collected after admission. Additionally, we did not include patients who developed AKI without COVID and were unable to determine whether COVID-AKI has unique proteomic markers compared to other forms of sepsis-AKI. Thus, our AKI cases may be biased towards less severe presentations. Second, since kidney injury is usually not an isolated phenomenon in critically ill patients, the protein expression changes observed may have been partially due to damage to other organs, such as the lung, liver, and heart. However, we accounted for non-kidney damage by adjusting for the highest level of ventilatory support and thus our results are likely a reflection of kidney injury. Specifically, our results do show the importance of crosstalk between the cardiac system and the kidneys. In addition, we did not include proteomic measurements from urine specimens and thus it is unclear whether poor filtration or resorption of proteins plays a role in peripheral blood protein concentrations. For example, poor resorption of cystatin-C in the setting of AKI may have led to the increased peripheral blood cystatin-C that we report. Our study was adequately powered to detect effect sizes of greater than or equal to 1.6. Also, since we enrolled patients only from March-October 2020, we cannot generalize our findings to other COVID-19 variants and time periods. Although we adjusted our regression models for history of CKD, it is possible that unmeasured confounding due to preexisting impaired kidney function has not completely been controlled our in our analyses. Another big limitation of our study is the inability to significantly distinguish between effects of impaired filtration and true pathogenic differences in protein levels without having to conduct additional validation tests.

We were not able to exclude individuals who were lost to follow-up or died because the data was extracted from an institutional EHR. Some patients accessed care at other hospitals after discharge. This remains a limitation as well. Finally, our cohort did not include autopsy or kidney biopsy specimens. Histopathological analysis of kidney specimens is necessary to determine the mechanism of AKI and whether viral particles are present in the kidney.

In conclusion, we provide, to the best of our knowledge, the first comprehensive characterization of the plasma proteome of AKI and long term eGFR decline in hospitalized COVID-19 patients. Our results suggest in the setting of COVID-AKI and post-discharge kidney dysfunction there is evidence of tubular damage in the peripheral blood but that in the acute setting,

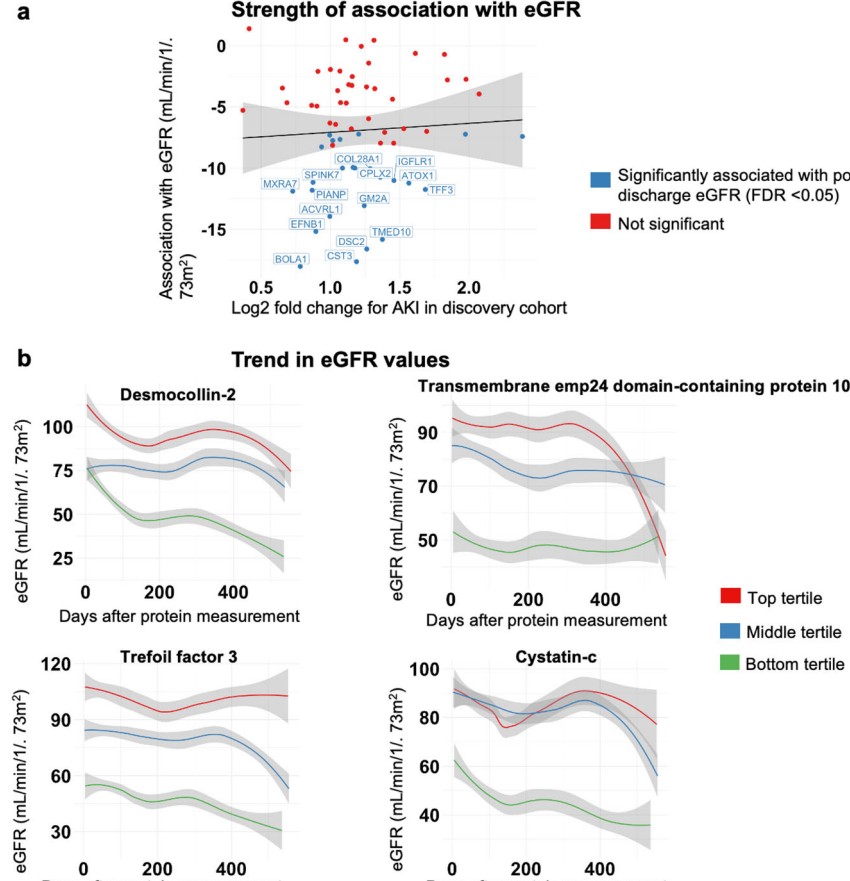

**Fig. 5 Proteomic characterization of long-term eGFR decline. a** Comparison of strengths of association with AKI and long term eGFR for proteins associated with Acute Kidney Injury (AKI) in both the discovery and validation cohorts ($n = 62$). **b** Trend in estimated glomerular filtration rate (eGFR) values separated by protein expression for tertiles for proteins most significantly (by *P*-value) associated with eGFR trend. Source data for **a**. Proteomic characterization of post-discharge kidney function is derived from Supplementary Data 1 (sheet "Table 3") and is created using this code: Code for Fig. 5 on GitHub.

several factors including hemodynamic disturbance and myo-cardial injury also play a role.

## Data availability

This research was reviewed and approved by the Icahn School of Medicine at Mount Sinai Program for the Protection of Human Subjects (PPHS) under study number 20-00341. The clinical data tables and the analysis data are available in Synapse repository syn35874390. Synapse can be accessed here https://www.synapse.org/#!Synapse:syn35874390.

## Code availability

Code is available at our GitHub repository Nadkarni-Lab: aki_covid_proteomics here: https://github.com/Nadkarni-Lab/aki_covid_proteomics/releases/tag/v0.1.2. The code[52] has been made citable with DOI: and target link: https://zenodo.org/badge/latestdoi/436359832. Code to generate Fig. 2, Fig. 3 and Fig. 5 are in the GitHub code repository within the folder, code_and_data_for_figures_in_paper.

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

## Acknowledgements

We would like to thank the scientists at Somalogic for their assistance with technical and scientific questions about the assay. E.U.A., G.N.N., J.C.H. and S.G.C. are partially funded by R01 DK118222. G.N.N. also is supported by R01DK127139, R01HL155915 and R56DK126930.

## Author contributions

Conceptualization: I.P., G.N.N., A.W.C., S.Z., N.B., E.S., E.K., L.C., S.C., B.M., S.D. Methodology: I.P., P.J., R.T., K.C., S.A., A.V., S.J., M.P., R.O.H., S.K., Si.Zh., E.U.A., F.F.G., M.S.-F. Investigation: I.P., P.J., R.T., E.K., Sh.Zh., S.J., K.C., A.K., S.G., H.X., J.H., M.P., K.A., C.B., K.N., R.O.H. Visualization: I.P., P.J. Funding acquisition: G.N.N., A.W.C., B.S.G., M.M., N.B. Supervision: G.N.N., A.W.C., B.S.G., M.M., S.K., N.B., E.S., J.C.H. Writing: I.P., P.J., G.N.N., S.K. Revisions: P.J., I.P., G.N.N.

## Competing interests

G.N.N. and S.G.C. reports grants, personal fees, and non-financial support from Renalytix. G.N.N. reports non-financial support from Pensieve Health, personal fees from AstraZeneca, personal fees from BioVie, personal fees from GLG Consulting, personal fees from Siemens Healthineers from outside the submitted work. I.P. receives personal fees from Character Biosciences. The other authors declare no competing interests.

## Additional information

[1]Department of Medicine, Stanford University, Stanford, CA, USA. [2]The Charles Bronfman Institute for Personalized Medicine (CBIPM), Division of Data Driven and Digital Medicine (D3M), Icahn School of Medicine at Mount Sinai, New York, NY, USA. [3]Lady Davis Institute, Jewish General Hospital, McGill University, Montreal, QC, Canada. [4]Department of Computer Science, Quantitative Life Sciences, McGill University, Montreal, QC, Canada. [5]Department of Epidemiology, Biostatistics and Occupational Health, McGill University, Montreal, QC, Canada. [6]The Precision Immunology Institute, Icahn School of Medicine at Mount Sinai, New York, NY, USA. [7]The Mount Sinai Clinical Intelligence Center (MSCIC), The Charles Bronfman Institute for Personalized Medicine (CBIPM), Icahn School of Medicine at Mount Sinai, New York, NY, USA. [8]Department of Genetics and Genomic Sciences, Icahn School of Medicine at Mount Sinai, New York, NY, USA. [9]Icahn Institute for Genomics and Multiscale Biology, Icahn School of Medicine at Mount Sinai, New York, NY, USA. [10]Department of Anesthesiology, Perioperative and Pain Medicine, Icahn School of Medicine at Mount Sinai, New York, NY, USA. [11]The Hasso Plattner Institute for Digital Health at Mount Sinai, Icahn School of Medicine at Mount Sinai, New York, NY, USA. [12]Clinical Informatics, CSIR-Institute of Genomics and Integrative Biology (CSIR-IGIB), New Delhi, India. [13]Division of Health Sciences and Technology, Harvard Medical School, Boston, MA, USA. [14]Human Immune Monitoring Center, Icahn School of Medicine at Mount Sinai, New York, NY, USA. [15]Department of Medicine, Division of Nephrology, Icahn School of Medicine at Mount Sinai, New York, NY, USA. [16]Department of Biostatistics, Icahn School of Medicine at Mount Sinai, New York, NY, USA. [17]Department of Oncological Sciences, Icahn School of Medicine at Mount Sinai, New York, NY, USA. [18]Department of Computer Science, McGill University, Montreal, QC, Canada. [19]Department of Human Genetics, McGill University, Montreal, QC, Canada. [20]Department of Twin Research, King's College London, London, GB, UK. [21]Data Science and Machine Learning, Character Biosciences, New York, NY, USA. [22]The Charles Bronfman Institute for Personalized Medicine (CBIPM), Icahn School of Medicine at Mount Sinai, New York, NY, USA. [23]The Pamela Sklar Division of Psychiatric Genomics, Icahn School of Medicine at Mount Sinai, New York, NY, USA. [24]These authors contributed equally: Ishan Paranjpe, Pushkala Jayaraman. ✉email: girish.nadkarni@mountsinai.org

