## [Peer Review File · Communications Medicine]

This manuscript has been previously reviewed at another Nature Portfolio journal. This document only contains reviewer comments and rebuttal letters for versions considered at Communications Medicine. Mentions of the other journal have been redacted.

REVIEWERS' COMMENTS:

Reviewer #1 (Remarks to the Author):

Authors have addressed further comments and questions. I have checked the similarity of the recent manuscript to that submitted to [redacted] before

Reviewer #4 (Remarks to the Author):

The authors now sufficiently acknowledge the limitation of the study and I do not have any further concerns. The paper clearly improved during the review process and I thank the authors for going through all this effort. Well done.

Reviewer #5 (Remarks to the Author):

The authors have addressed my concerns appropriately. I would like to highlight that the use of a hypergeometric test for assessing the replicability is not ideal though, because the significance may be (strongly) influenced by the correlation between the variables.

REVIEWERS' COMMENTS:

Reviewer #1 (Remarks to the Author).

1. Authors have addressed further comments and questions. I have checked the similarity of the recent manuscript to that submitted to [redacted] before

Response. The authors would like to thank the reviewer for their generous time and valuable inputs to make our manuscript better and improve the quality of our publication.

Reviewer #4 (Remarks to the Author).

1. The authors now sufficiently acknowledge the limitation of the study and I do not have any further concerns. The paper clearly improved during the review process, and I thank the authors for going through all this effort. Well done.

Response. The authors would like to thank the reviewer for their generous time and valuable inputs to make our manuscript better and improve the quality of our publication.

Reviewer #5 (Remarks to the Author).

1. The authors have addressed my concerns appropriately. I would like to highlight that the use of a hypergeometric test for assessing the replicability is not ideal though, because the significance may be (strongly) influenced by the correlation between the variables.

Response. The authors would like to thank the reviewer for their observations.

To address replicability, we conducted tests to measure agreement using Cohen's Kappa¹ metric. This kappa statistic showed a positive agreement (0.257) between the lists with 91% agreement result at a p-value <0.001.

In addition, we also conducted permutation testing using *CompareList* function from the *OrderedList* R package using 10000 permutations. We ran two instances of permutation testing ordering by p-value and log fold change. In both causes, we found significant (P <0.001) overlap.

List comparison	ordered by p value	Ordered by log fold change
Assessing similarity of	top ranks	top ranks
Length of lists	4429	4429
Number of random samples	10000	10000
Lists are more alike in direct order		
Chosen regularization parameter	alpha = 0.058 (200 genes)	alpha = 0.058 (200 genes)
Weighted overlap score	129.5	177.1
Significance of similarity	p-value < 0.001	p-value < 0.001
Score percentage for common entries	95	95
Entries contributing score percentage	50	53

Because of the comparable results between the permutation and hypergeometric testing for the significant proteins, we believe that that the results from these tests support our findings from the analysis as well.

References.

- 1 McHugh, M. L. Interrater reliability: the kappa statistic. *Biochem Med (Zagreb)* **22**, 276-282 (2012).

COMMENTS FROM REVIEWER 5

The problem I have tried to highlight with my comment is that the “hypergeometric test”, which tests for significance of the overlap of significant marker lists, is based on the assumption that these markers (e.g. proteins) are statistically independent from each other.

With proteins, this is not the case, because of their sometimes strong co-regulation, and the underlying co-expression networks. Therefore, it can (and does) happen that significant lists significantly overlap, because they relate to a set of highly correlated proteins.

To resolve this, one can permute the clinical outcome, and re-compute the list of P-values. This preserves the correlation structure between proteins, and the subsequent analysis of overlap will not be biased.

The authors did not address this in their revision, because it appears they performed permutation analysis directly on the list of P-values, which would not address the actual issue.

Despite this, it can be considered that the hypergeometric testing procedure is, although incorrect, relatively common practice, due to its simplicity. Unfortunately, it is difficult to judge how strong the bias is unless the overlap is very strong, which, however, I don't think it is. The description of significance is also not entirely consistent, as the authors state <0.05 in the manuscript and <0.001 in their first rebuttal letter.

I was also wondering where the procedure to assess this is actually described in the methods. Unfortunately, I was not able to find it.

Therefore, I am afraid, I could not with confidence say that the issue has been resolved completely.

COMMENTS: REVIEWER 5

The problem I have tried to highlight with my comment is that the “hypergeometric test”, which tests for significance of the overlap of significant marker lists, is based on the assumption that these markers (e.g. proteins) are statistically independent from each other.

With proteins, this is not the case, because of their sometimes strong co-regulation, and the underlying co-expression networks. Therefore, it can (and does) happen that significant lists significantly overlap, because they relate to a set of highly correlated proteins.

To resolve this, one can permute the clinical outcome, and re-compute the list of P-values. This preserves the correlation structure between proteins, and the subsequent analysis of overlap will not be biased.

The authors did not address this in their revision, because it appears they performed permutation analysis directly on the list of P-values, which would not address the actual issue.

Despite this, it can be considered that the hypergeometric testing procedure is, although incorrect, relatively common practice, due to its simplicity. Unfortunately, it is difficult to judge how strong the bias is unless the overlap is very strong, which, however, I don't think it is. The description of significance is also not entirely consistent, as the authors state <0.05 in the manuscript and <0.001 in their first rebuttal letter.

I was also wondering where the procedure to assess this is actually described in the methods. Unfortunately, I was not able to find it.

Therefore, I am afraid, I could not with confidence say that the issue has been resolved completely.

>> The authors thank the reviewer for their detailed suggestion to test our result. We now understand the request by the reviewer and have implemented “permutation testing analysis” for the confirmation of their significant genes on both, the discovery and validation cohorts. The authors performed the following steps for this analysis

1. For the discovery and validation cohorts:
 - a. We permuted the AKI outcome variable for all samples to create 100000 instances of permutations.
 - b. For each instance:
 - i. We ran the limma regression model and identified differentially expressed genes using the permuted outcome.
 - c. For each gene, across 100000 instances, we created the empirical null distribution of t statistics
 - d. We then compared this empirical null distribution against the true AKI outcome variable to compute empirical P values defined as:
 - i. We identified the number of instances out of the 100,000 where the absolute value of the t statistics from the empirical null was greater than or equal to the t statistic value from the truth file.
 - e. Empirical P values were adjusted for multiple comparisons using the FDR.

Results:

2. **Intersecting the common genes between discovery and validation cohorts:**
 - a. We interested the DE genes from the discovery and validation cohorts and found 272 statistically significant genes.
 - b.
3. We found that all 62 genes identified in the original analysis were also had an empirical P value <0.05 after the permutation testing.

We have attached the results from our permutation testing that list the p-value and the FDR for each of the 62 genes for each cohort (table1 below).

In addition, we also performed hypergeometric test and measured the Cohen’s Kappa once again **Results:** The hypergeometric test for overlap between the significant proteins in the validation and discovery cohorts was also significant (P = 2.133E-159). Additionally, the Cohen’s Kappa between the two protein lists based on P value was 0.501.

Table1: Comparison of the p-values after permutation testing for the 62 genes.

Target	UniProt	Entrez Gene Symbol	full gene name	Discovery Cohort			Validation Cohort		
				tstats	Count >= ground truth	P Val	tstats	Count >= ground truth	P Val
ALK-1	P37023	ACVRL1	Serine/threonine-protein kinase receptor R3	12.72395606	0	0	4.598939848	6	6.00E-05
Angiopoietin-2	O15123	ANGPT2	Angiopoietin-2	5.778974815	0	0	2.714652605	738	0.00738
Apo F	Q13790	APOF	Apolipoprotein F	7.86380747	0	0	3.993481888	18	0.00018
ASGR1	P07306	ASGR1	Asialoglycoprotein receptor 1	7.441806725	0	0	4.684083378	4	4.00E-05

ATOX1	O00244	ATOX1	Copper transport protein ATOX1	14.79760184	0	0	5.281096398	0	0
BOLA1	Q9Y3E2	BOLA1	BolA-like protein 1	12.16794545	0	0	5.297265907	0	0
Carbonic anhydrase III	P07451	CA3	Carbonic anhydrase 3	7.578252754	0	0	3.203515231	186	0.00186
CD59	P13987	CD59	CD59 glycoprotein	9.899959182	0	0	4.820947452	0	0
SCG1	P05060	CHGB	Secretogranin-1	14.08173049	0	0	5.211742713	0	0
COL	P04118	CLPS	Colipase	8.867440321	0	0	3.301937785	178	0.00178
COSA1	Q2UY09	COL28A1	Collagen alpha-1(XXVIII) chain	13.36327142	0	0	5.670013394	0	0
CPLX2	Q6PUV4	CPLX2	Complexin-2	12.28690253	0	0	4.738531632	2	2.00E-05
CRK	P46108	CRK	Adapter molecule crk	8.973633925	0	0	4.176349475	1	1.00E-05
Cystatin C	P01034	CST3	Cystatin-C	15.02757845	0	0	6.155259359	0	0
CYTD	P28325	CST5	Cystatin-D	7.48454853	0	0	3.195215413	171	0.00171
Chymotrypsin	P17538	CTRB1	Chymotrypsinogen B	6.163949729	0	0	4.155183319	9	9.00E-05
ACBP	P07108	DBI	Acyl-CoA-binding protein	15.73168536	0	0	5.350641282	0	0
Discoidin domain receptor 2	Q16832	DDR2	Discoidin domain-containing receptor 2	9.962324573	0	0	4.494363526	4	4.00E-05
EGFL9	Q6UY11	DLK2	Protein delta homolog 2	8.224380504	0	0	3.931254468	17	0.00017
DJB12	Q9NXW2	DNAJB12	DnaJ homolog subfamily B member 12	8.656881472	0	0	3.825087274	17	0.00017
DSC2	Q02487	DSC2	Desmocollin-2	14.53045114	0	0	5.495867273	0	0
EFNB1	P98172	EFNB1	Ephrin-B1	14.37502479	0	0	7.396864761	0	0
EFNB2	P52799	EFNB2	Ephrin-B2	7.965874595	0	0	3.758587057	31	0.00031
Epithelial cell kinase	P29317	EPHA2	Ephrin type-A receptor 2	10.04532675	0	0	5.105784226	1	1.00E-05
FABP	P05413	FABP3	Fatty acid-binding protein, heart	9.973958285	0	0	3.936719921	14	0.00014
SAP3	P17900	GM2A	Ganglioside GM2 activator	14.7983567	0	0	6.989924786	0	0
HDGF	P51858	HDGF	Hepatoma-derived growth factor	5.145341503	0	0	2.047193655	4259	0.04259
TM149	Q9H665	IGFLR1	IGF-like family receptor 1	11.37743904	0	0	3.906606358	11	0.00011
IgG4, Kappa	P01861	IGHG4	Ig gamma-4, Kappa	7.225380325	0	0	5.614906313	0	0
Lipocalin 2	P80188	LCN2	Neutrophil gelatinase-associated lipocalin	10.16756756	0	0	4.330712248	3	3.00E-05

Myoglobin	P02144	MB	Myoglobin	10.42674 542	0	0	2.9988 25565	376	0.00376
MXRA7	P84157	MXRA7	Matrix-remodeling-associated protein 7	10.15403 388	0	0	6.0790 63804	0	0
Myosin light chain 1	P08590	MYL3	Myosin light chain 3	8.235074 86	0	0	2.8351 85357	525	0.00525
DAN	P41271	NBL1	Neuroblastoma suppressor of tumorigenicity 1	14.10672 958	0	0	6.7404 2808	0	0
N-terminal pro-BNP	P16860	NPPB	N-terminal pro-BNP	5.300911 758	0	0	2.2253 22217	2767	0.02767
PCDGA	Q9Y5H3	PCDHGA10	Protocadherin gamma-A10	8.415558 273	0	0	3.7403 95269	47	0.00047
PIANP	Q8IYJ0	PIANP	PILR alpha-associated neural protein	10.88352 279	0	0	2.1592 96383	3158	0.03158
LIPR1	P54315	PNLIPRP1	Inactive pancreatic lipase-related protein 1	8.098872 905	0	0	3.5796 57094	47	0.00047
Trypsin	P07477	PRSS1	Trypsin-1	7.520788 226	0	0	3.8100 1249	25	0.00025
Trypsin 2	P07478	PRSS2	Trypsin-2	11.32386 081	0	0	5.2271 34613	1	1.00E-05
TRY3	P35030	PRSS3	Trypsin-3	8.967646 529	0	0	4.8371 18072	0	0
PTH	P01270	PTH	Parathyroid hormone	7.878645 874	0	0	3.5205 52176	66	0.00066
PXDN	Q92626	PXDN	Peroxidasin homolog	13.41575 623	0	0	4.8475 4534	0	0
PSP	P05451	REG1A	Lithostathine-1-alpha	14.17458 715	0	0	5.2462 47684	1	1.00E-05
PAP1	Q06141	REG3A	Regenerating islet-derived protein 3-alpha	11.24273 317	0	0	4.9084 67627	1	1.00E-05
RNase 1	P07998	RNASE1	Ribonuclease pancreatic	14.35865 859	0	0	4.1252 90322	6	6.00E-05
RNAS6	Q93091	RNASE6	Ribonuclease K6	11.97576 266	0	0	3.9395 331	9	9.00E-05
ROR2	Q01974	ROR2	Tyrosine-protein kinase transmembrane receptor ROR2	12.93705 739	0	0	3.9644 55547	20	2.00E-04
SELW	P63302	SEPW1	Selenoprotein W	8.277672 661	0	0	3.4432 86183	79	0.00079
ISK7	P58062	SPINK7	Serine protease inhibitor Kazal-type 7	9.072302 857	0	0	5.3149 38234	0	0
Osteopontin	P10451	SPP1	Osteopontin	9.928970 619	0	0	3.3955 87223	122	0.00122
SRCA	Q86TD4	SRL	Sarcalumenin	14.21867 87	0	0	4.4331 93794	5	5.00E-05
STMN3	Q9NZ72	STMN3	Stathmin-3	5.909098 577	1	0.044 96	5.0231 11972	1	1.00E-05
SUMO2	P61956	SUMO2	Small ubiquitin-related modifier 2	15.71934 188	0	0	4.1501 81565	4	4.00E-05

TAGL	Q01995	TAGLN	Transgelin	11.65964 555	0	0	4.8851 63211	0	0
Trefoil factor 2	Q03403	TFF2	Trefoil factor 2	8.487414 97	0	0	4.9048 98761	3	3.00E-05
TFF3	Q07654	TFF3	Trefoil factor 3	15.02572 999	0	0	4.4233 27069	5	5.00E-05
TMEDA	P49755	TMED10	Transmembrane emp24 domain-containing protein 10	15.24531 038	0	0	6.5152 30398	0	0
TAJ	Q9NS68	TNFRSF19	Tumor necrosis factor receptor superfamily member 19	13.14367 736	0	0	5.0150 33288	0	0
TNF sR-I	P19438	TNFRSF1A	Tumor necrosis factor receptor superfamily member 1A	9.946876 065	0	0	5.0151 46035	0	0
Troponin T	P45379	TNNT2	Troponin T, cardiac muscle	11.49971 959	0	0	3.1976 61947	166	0.00166
TITIN	Q8WZ42	TTN	Titin	10.99864 194	0	0	4.9473 30952	3	3.00E-05

REVIEWERS' COMMENTS:

Reviewer #5 (Remarks to the Author):

The authors have now implemented a permutation approach as part of their analysis workflow.

However, it appears the main issue of my concern has not been understood well.

The issue is that the *overlap* of the P-value lists of discovery and validation cohorts can arise by chance (e.g. if you have two relatively large lists of proteins with $FDR < 0.05$, and the overlap relates to a smaller, highly correlated set of proteins).

To test this, the permutation needs to target the overlap of the p-value lists, i.e. it would need to be tested in how many of the permutations, there is an overlap of at least 62 genes between the training and the validation data. It has to be considered though that when FDR control is part of the workflow, during permutation one is likely to observe a far lower number of significant proteins, and thus overlap. Instead, one would likely take the same number of top significant proteins as in the original analysis, and then test the overlap.

In any case, I would now suggest accepting as is, with the following modifications:

- add to the methods and results that FDR was performed on training and validation data with respect to all proteins measured
- modify the text of page 15 accordingly, which still states that the 62 proteins only show nominal significance in the validation data
- . state the overall number of significant proteins after FDR in the validation data

REVIEWERS' COMMENTS:

Reviewer #5 (Remarks to the Author):

The authors have now implemented a permutation approach as part of their analysis workflow.

However, it appears the main issue of my concern has not been understood well.

The issue is that the *overlap* of the P-value lists of discovery and validation cohorts can arise by chance (e.g. if you have two relatively large lists of proteins with $FDR < 0.05$, and the overlap relates to a smaller, highly correlated set of proteins).

To test this, the permutation needs to target the overlap of the p-value lists, i.e. it would need to be tested in how many of the permutations, there is an overlap of at least 62 genes between the training and the validation data. It has to be considered though that when FDR control is part of the workflow, during permutation one is likely to observe a far lower number of significant proteins, and thus overlap. Instead, one would likely take the same number of top significant proteins as in the original analysis, and then test the overlap.

In any case, I would now suggest accepting as is, with the following modifications:

- add to the methods and results that FDR was performed on training and validation data with respect to all proteins measured
- modify the text of page 15 accordingly, which still states that the 62 proteins only show nominal significance in the validation data
- . state the overall number of significant proteins after FDR in the validation data

>> We thank the reviewer for their patience and their feedback through this session. We have made the suggested edits in the manuscript. The suggested line has been added to the **Methods and Results section**.

- a. - add to the methods and results that FDR was performed on training and validation data with respect to all proteins measured

Differential expression analysis for prevalent AKI

Using data from the AKI cohort, \log_2 transformed normalized protein values were modelled using multivariable linear regression in the Limma framework³⁹. Models were adjusted for age, sex, history of chronic kidney disease (CKD), and supplemental oxygen requirement (0,1, or 2 [see above]) at the time of specimen collection. P-values were adjusted using the Benjamin-Hochberg procedure to control the false discovery rate (FDR) at 5%. FDR was performed on discovery and validation data with respect to all proteins measured.

RESULTS

Discovery and Validation Cohort Overview

To discover proteins associated with COVID-AKI, we enrolled a prospective cohort of patients hospitalized with COVID-19 admitted between March 24, 2020 and August 26, 2020 into a biobank as previously described³⁴. Cases were defined as patients who developed AKI (stage 2 or 3) during their hospital admission and controls included all other patients (**Fig 1**). Characteristics of cases and controls in the discovery cohort are provided in **Supplementary Data 1 (sheet “Table 1”)**. Patients who developed AKI (stage 2 or 3) had a greater prevalence of diabetes (42% vs 22%, $p < 0.001$), and chronic kidney disease (31% vs 5%, $p < 0.001$) and more frequently required intubation (46% vs 11%, $p < 0.001$). Patients who developed AKI (stage 2 or 3) also had a significantly lower minimum systolic blood pressure (104 vs 110, $p < 0.001$), greater maximum pulse (106 vs 94, $p < 0.001$), white blood cell count (12.9 vs 8.8, $p < 0.001$), ferritin (2210 vs 1030 $p < 0.001$), and frequency of vasopressor use (48% vs 14%, $p < 0.001$). We validated proteomic associations in an external cohort from Quebec, Canada. Characteristics of the validation cohort are provided in **Supplementary Table 1 (See**

Supplementary Information). In the validation cohort, compared to controls, AKI (stage 2 or 3) cases have a significantly higher prevalence of CKD (29% vs 11%, $p = 0.01$) and a higher rate of intubation at the time of sample collection (49% vs 13%). **FDR was performed on discovery and validation data with respect to all proteins measured.**

- modify the text of page 15 accordingly, which still states that the 62 proteins only show nominal significance in the validation data

. state the overall number of significant proteins after FDR in the validation data

>> Thank you for the suggestions. We have made the following modifications:

Validation of AKI-associated proteins

We then performed an external validation of AKI-associated proteins in a prospective biobank cohort from Quebec, Canada. **443 proteins in the discovery cohort are significantly associated with AKI (FDR adjusted $P < 0.05$) while 71 proteins in the validation cohort are significantly associated with AKI ($p < 0.05$). Of the proteins significantly associated with AKI in the discovery cohort, 62 are also associated with AKI in the validation cohort ($p < 0.05$, See Supplementary Data 1 (sheet "Table 2").** The hypergeometric test for overlap between the significant proteins in the validation and discovery cohorts was also significant ($P = 2.133E-159$). Additionally, the Cohen's Kappa between the two protein lists based on P value was 0.501. **All validated proteins associate with an increased risk of AKI with nominal significance.** The correlation of fold changes of validated proteins in the discovery and validation cohort show a Pearson correlation score of 0.71 (**Fig 2b**). The 62-protein signature distinctly separate AKI cases from cohorts in the

discovery cohort (**Fig 2c**). To assess how many of our candidate proteins had orthogonal evidence for target specificity, we sought to identify how many of our proteins contained reported plasma protein quantitative trait loci (pQTL) associations from a recent publication by Ferkingstad *et al.* (Nat Genet, 2021). Of the 62 AKI associated proteins, 45 have both cis and trans pQTLs, 14 have only trans pQTLs, and 2 had cis pQTLs (**See Supplementary Figure 1, Supplementary Information**). Protein-protein interaction (PPI) network analysis reveal enrichment of several highly connected proteins, including LCN2 (alternative name: NGAL), REG3A, and MB (**Fig 4a**). The AKI-associated protein network also includes a cluster of cardiac structural proteins (**Fig 4b**), TNNT2, TTN, MYL3, SRL, and NPPB (alternative name: BNP).